# The impact of restricted provision of publicly funded elective hip and knee joints replacement during the COVID-19 pandemic in England

Chris M. Penfold[1,2,3]*, Ashley W. Blom[1,2], Maria Theresa Redaniel[3,4], Tim Jones[3,4], Emily Eyles[3,4], Tim Keen[5], Andrew Elliott[5], Andrew Judge[1,2]

1 Musculoskeletal Research Unit, Translational Health Sciences, Bristol Medical School, 1st Floor Learning & Research Building, Southmead Hospital, Bristol, United Kingdom, 2 National Institute for Health Research Bristol Biomedical Research Centre, University Hospitals Bristol and Weston NHS Foundation Trust and University of Bristol, Bristol, United Kingdom, 3 The National Institute for Health and Care Research Applied Research Collaboration West (NIHR ARC West) at University Hospitals Bristol and Weston NHS Foundation Trust, Bristol, United Kingdom, 4 Population Health Sciences, Bristol Medical School, University of Bristol, Bristol, United Kingdom, 5 North Bristol NHS Trust, Southmead Hospital, Westbury-on-Trym, Bristol, United Kingdom

* chris.penfold@bristol.ac.uk

## Abstract

### Aims

Elective hip and knee replacement operations were suspended in April 2020 due to the COVID-19 pandemic. The impact of this suspension and continued disruption to the delivery of joint replacement surgery is still emerging. We describe the impact of the pandemic on the provision of publicly funded elective hip and knee replacement surgery at one teaching hospital in England and on which patients had surgery.

### Methods

We included all elective primary and revision hip and knee replacements performed at one hospital between January 2016 and June 2021. Using data for the years 2016–2019, we estimated the expected number of operations and beds occupied per month in January 2020 to June 2021 using time series linear models (adjusting for season and trend). We compared the predictions with the real data for January 2020 to June 2021 to assess the impact of the pandemic on the provision of elective hip and knee replacements. We compared the length of stay and characteristics (age, gender, number of comorbidities, index of multiple deprivation) of patients who had surgery before the pandemic with those who had surgery during the pandemic.

### Results

We included 6,964 elective primary and revision hip and knee replacements between January 2016 and June 2021. Between January 2020 and June 2021 primary hip replacement volume was 59% of predicted, and 47% for primary knee replacements. Revision hip

**Data Availability Statement:** Access to the data in this study required permission from North Bristol

NHS Trust. A request for access to data may be made to North Bristol NHS Trust, www.nbt.nhs.uk.

**Funding:** This study was funded by the HDRUK Better Care Partnership (#6.12). This research was supported by the NIHR Biomedical Research Centre at University Hospitals Bristol and Weston NHS Foundation Trust and the University of Bristol and by the National Institute for Health Research (NIHR) Applied Research Collaboration West (NIHR ARC West). The views expressed in this article are those of the author(s) and not necessarily those of the NIHR or the Department of Health and Social Care. The funders had no role in study design, data collection and analysis, decision to publish, or preparation of the manuscript.

**Competing interests:** I have read the journal's policy and the authors of this manuscript have the following competing interests: AB (Stryker) has received research and other financial support from companies or suppliers outside the submitted work. AJ declares advisory board positions with receipt of fees (Anthera Pharmaceuticals, INC.) and paid consultancy work (Freshfields Bruckhaus Deringer) for companies outside the submitted work. All other authors declare no competing interests. This does not alter our adherence to PLOS ONE policies on sharing data and materials.

replacement volume was 77% of predicted, and 42% for revision knee replacement. Median length of stay was one day shorter for primary (4 vs 3 days) and revision (6 vs 5 days) operations during the pandemic compared with before. Patients operated on during the pandemic were younger and had slightly more comorbidities than those operated on before the pandemic.

## Conclusions

The restricted provision of elective hip and knee replacements during the COVID-19 pandemic changed the patient casemix, but did not introduce new inequalities in access to these operations. Patients were younger, had more comorbidities, and stayed in hospital for less time than those treated before the pandemic. Approximately half the number of operations were performed during the pandemic than would have been expected and the effect was greatest for revision knee replacements.

## Introduction

The population health burden of osteoarthritis (OA) is significant and growing worldwide [1]. Elective hip and knee replacements are two of the most commonly performed operations in the UK to relieve the pain and functional limitations of OA (an indication for surgery in more than 90% cases) [2–4]. They are effective, with approximately 90% of patients achieving clinically meaningful improvements in pain and function [5], have very low rates of postoperative mortality [6, 7] and typical 10-year revision rates of less than 5% [2]. The mean age of patients at the time of their primary surgery is 68 years (hips) and 69 years (knees), and the proportion female is 60% for hip and 56% for knee replacements [2]. Primary hip and knee replacements use implants to replace diseased or worn out hip and knee joints. Revision operations are performed to add, remove or modify one or more components of a hip/knee replacement or to treat a periprosthetic infection. The lifetime risk of hip replacement is 11.6% for women and 7.1% for men, and for knee replacement 10.8% for women and 8.1% for men [8]. The mean length of hospital stay after primary hip and knee replacements is approximately three days [9], and is approximately 1–1.5 days longer for revision operations [10].

Seasonal spikes in hospital admissions, including for example a five percentage point increase in average bed occupancy rates in winter 2015/2016 compared with summer, have been a priority challenge for the National Health Service (NHS) for a number of years [11]. During peak demand in 2016/2017, one third of trust Accidents and Emergencies departments in England declared that hospital performance/patient safety may be compromised [12]. Between 2011/2012 and 2018, more than £3billion was provided at short notice to support NHS services over winter [13]. Management plans have been extended before the COVID-19 pandemic to include increasing bed occupancy above operating limits (85%) and cancelling elective operations [14].

In recent years, NHS England have advised wide-spread cancellations of elective operations on two occasions. The first was during the 2017/2018 winter flu season when NHS England advised hospitals to suspend elective operations throughout January 2018 [15]. This resulted in January-March 2018 recording the highest number of people waiting >18 weeks for their NHS treatment since quarterly records began (1994) [16]. The second was due to the Covid-19 pandemic. All elective operations in the NHS were suspended from 15th April 2020 [17],

initially for 'at least three months'. In reality, many hospitals had begun cancelling these operations earlier in order to increase their capacity to treat adult COVID-19 patients [18].

The impact of the COVID-19 pandemic on the delivery of elective operations is still emerging as the pandemic continues to disrupt care. In this paper we describe the impact of the COVID-19 pandemic on elective hip and knee replacements at one teaching hospital in England.

Using data from one NHS Trust we will:

1. Compare the observed volume of elective primary and revision hip and knee replacements performed during the COVID-19 pandemic with those predicted from historic activity

2. Identify variation in the types of patients receiving elective primary and revision hip and knee replacements during the COVID-19 pandemic and any inequalities with respect to area-level deprivation

## Methods

### Setting

The setting for this study was one large (>900 beds) teaching hospital in Bristol, UK. Before 2020, a large number of primary hip (n~650/year) and knee (n~530/year) replacement operations were performed at this hospital compared with the national average (hip: approximately 240/year; knee: approximately 270/year) and 25–40% of the total number performed in the city [19].

### Ethical approval

We were provided with pseudonymised hospital admissions data from the NHS Trust under the NIHR ARC West Partnership Agreement. The project received ethical approval from the University of Bristol Faculty of Health Sciences ethical review board on 3rd November 2020 (ref# 109024).

### Study sample

The Trust provided data from their electronic medical records on all NHS funded admissions for elective hip and knee primary replacement and revision operations performed at the Trust between 1st January 2016 and 30th June 2021. We included all patients who received a primary or revision hip or knee replacement operation defined by a combination of OPCS4 procedure codes and surgical site codes (S1 Table). We excluded patients who were non-elective admissions and for whom we could not unambiguously determine the site of surgery or type of operation.

### The impact of restricted surgical capacity

**Study timeline.** We designated the period from 15th April 2020, the official suspension of elective operations due to the COVID-19 pandemic, until the end of our study as "COVID-19 restrictions". National guidance for patients undergoing elective procedures needing anaesthesia, produced in June 2020 by the National Institute for Health and Care Excellence, were that patients should socially distance for 14 days before admission, have a SARS-CoV-2 test three days before admission, and then self-isolate from the test until admission [20]. The "COVID-19 restrictions" period was preceded by hospital reorganisations to increase capacity for acute and critical care patients, meaning that elective activity was declining before officially being suspended. We have therefore designated the period from 1st January to 15th April 2020 as

"COVID-19 preparation", in recognition of the reorganisation of the provision of hospital treatments and preparation for the introduction of COVID-19 restrictions. We designated the period 1$^{st}$ January 2016 to 31$^{st}$ December 2019 as "Pre COVID-19".

**Change in patients being treated.** We compared the characteristics of patients treated during the COVID-19 preparation and restriction periods with those treated during the 'pre COVID-19' period to understand which patients were prioritised for treatment during these periods of restricted activity.

**Hospital activity.** We described the number of eligible operations performed (volume of operations), the total number of beds occupied by these patients and the median number of days these patients stayed in hospital (length of stay) per calendar month.

**Patient characteristics.** We described the patients treated in each of the study periods using the following characteristics:

- Median age at the time of their operation in years

- Number of comorbidities, an indicator of more complex clinical management [21] and predictor of risk of in-hospital mortality after joint replacement [22]

○ We counted the number of pre-existing conditions (comorbidities) in patients from those included in the Charlson Comorbidity Index (CCI) [23]. The CCI includes 17 conditions, which were recorded at the time of their operation using ICD-10 codes (see S2 Table for a full list of ICD-10 codes).

○ Number of comorbidities was categorised into '0', '1', '2' and '3 or more'

○ We calculated the proportion of patients operated on each month in each CCI category

- Gender

- Index of Multiple Deprivation (IMD) derived from the patient's home address, a measure of relative levels of deprivation in small areas (lower-level super output areas) in England [24]

○ We categorised IMD into quintiles 'Least deprived' through to 'Most deprived'

○ We calculated the proportion of patients operated on each month in each IMD quintile

## Statistical analysis

**Time-series modelling of elective arthroplasties.** To estimate the reduction in surgical activity during the COVID-19 preparation and restriction periods we used time-series linear regression models, including terms for seasonality and overall trend, to predict the expected monthly operation volume and bed occupancy during these periods using all available preceding activity (from 01/01/2016 to 31/12/2019). We then compared the expected and observed operation volume and bed occupancy during these periods to estimate the reduction in surgical activity. We predicted monthly operation volume and bed occupancy by primary and revision operations, and by hip and knee replacement.

**Comparison of patient demographics and clinical characteristics.** We compared the demographics and clinical characteristics of patients who received operations during the pre-COVID 19 period with those treated either in the COVID-19 preparation or COVID-19 restrictions periods. Since numbers in some categories were very low we combined the COVID-19 preparation and COVID-19 restrictions periods. We compared characteristics using the Wilcoxon rank sum test for non-normal continuous variables (age and length of hospital stay) and Pearson's chi-square test and Fishers exact test for categorical variables. We also

used Poisson regression models (adjusted for age, gender, IMD and number of comorbidities) to compare length of hospital stay.

**Sensitivity analyses.** The four years of pre-COVID-19 surgical activity we used to predict surgical activity during the COVID-19 pandemic may have incorporated organisational changes in the delivery of elective joint replacements. Inclusion of these data, while beneficial in modelling the seasonality of activity, may lead to biased predictions of surgical activity and bed occupancy. We therefore repeated our predictions of surgical activity and bed occupancy using a shorter Pre-COVID-19 period, from January 2018 to December 2019.

The longer pre-COVID 19 period used in our primary analyses may also include changes in the demographics and clinical characteristics of patients. We therefore repeated our comparison of these characteristics over the same shorter two year pre-COVID-19 period as described above.

Analyses were done in R version 4.1 [25] using the 'tidyverse' suite of packages [26] and the 'comorbidity' [27], and 'forecast' [28] packages.

## Results

Our study sample included 6,964 eligible operations performed between 1st January 2016 and 30th June 2021 at the Trust (S3 Table). Eight hundred and seventy-four (13%) were confirmed revision operations. In the 'Pre COVID-19' period 59% patients were female with a median age of 69. Fifty-five percent had no comorbidities and 13% had two or more. The most common comorbidities in this population were chronic pulmonary disease, diabetes without chronic complications, renal disease, and history of myocardial infarction (S4 Table). The proportion of patients increased as area-level deprivation decreased (most deprived 14%, least deprived 27%). The site of surgery was equally split between hips and knees.

### A comparison of observed versus predicted hospital activity

Two hundred and thirty operations were performed during the 'COVID-19 preparation' phase, of which 90% were confirmed primary and 10% confirmed revision operations. Since the official suspension of elective operations (15th April 2020) until the end of our study period, 690 operations had been performed (Fig 1), of which 85% were confirmed primary and 15% confirmed revision operations. Using time series modelling of surgical activity from January 2016 to December 2019, the predicted volume of elective primary and revision hip and knee replacements during COVID-19 preparation and restriction periods in the absence of restricted activity is 1,728 operations in total (Fig 2 and Table 1). The overall observed surgical volume was 53% of the predicted volume. The volume of knee replacement activity was more adversely affected by the COVID-19 restrictions than the volume of hip replacements, for both primary and revision operations. For primary joint replacements, the volume of knee replacements was 47.2% of predicted volume compared with 58.5% predicted volume for hip primaries. The volume of hip revision operations was 76.6% of predicted volume whereas knee revisions were only 41.7% of predicted volume. The predicted total bed occupancy during the COVID-19 preparation and restriction periods was 8,535 bed-days compared with observed occupancy of 4,393 bed-days (51.2%, Fig 3). For primary operations and hip revisions the proportions of observed to predicted bed occupancy were similar (primary hips: 56.9%, primary knees: 52.0%, revision hips: 57.9%), but were lower for revision knees (31.6%).

**Sensitivity analyses.** Predictions of surgical activity using a shorter pre-COVID time period (from January 2018 to December 2019) support our findings of substantially reduced observed surgical activity during than pandemic than predicted (Table 1). Predictions of primary joint replacement volume were higher than those from our main analyses, suggesting the

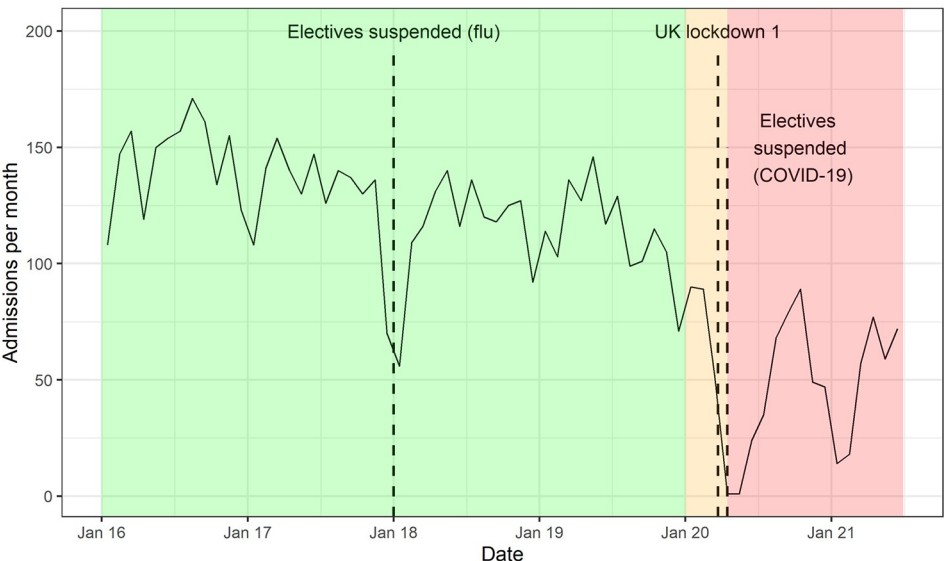

**Fig 1. Annotated time-series of elective total hip replacements and knee replacements performed per month, 2016 to 2021.** Note–background colours signify phases of the COVID-19 pandemic: green = 'Pre COVID-19', orange = 'COVID-19 preparation', red = 'COVID-19 restrictions'.

impact of the pandemic restrictions could have been larger (lower proportion of observed to predicted activity). Predictions of revision joint replacement volume were more variable. The prediction of revision knee procedure volume using the shorter pre-COVID-19 period was similar to our main analyses. But the prediction of revision hip procedure volume was much lower using the shorter pre-COVID-19 period compared with main analyses.

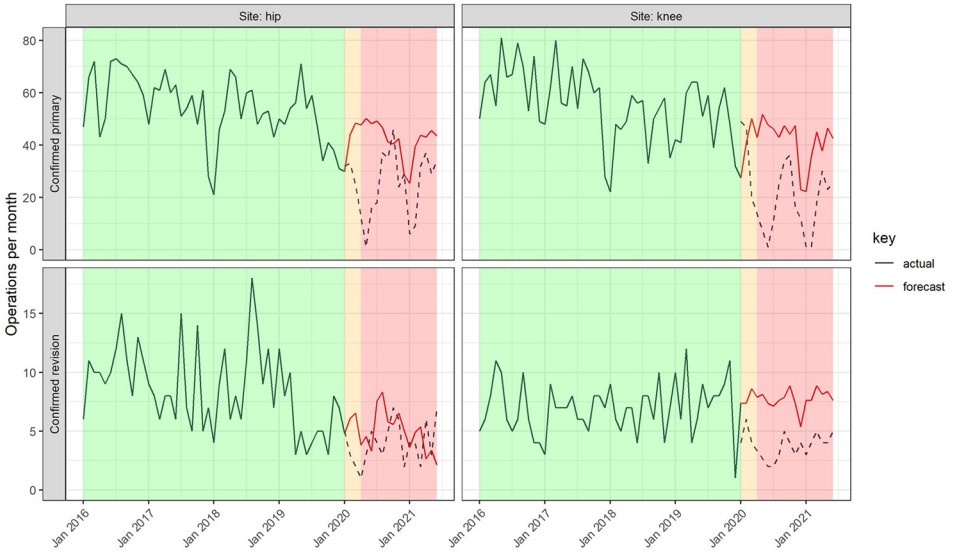

**Fig 2. A comparison of the observed volume of elective hip and knee replacements 2016–2019 (solid black) compared with the predicted volume (solid red) and observed volume (dashed black) between 1st January 2020 and 30th June 2021 per calendar month, by operation (primary or revision joint replacement operation) and location (hip or knee).** Note–background colours signify phases of the COVID-19 pandemic: green = 'Pre COVID-19', orange = 'COVID-19 preparation', red = 'COVID-19 restrictions'.

**Table 1. The observed volume of elective hip and knee replacements before (2016–2019 and 2018–2019) and during the COVID-19 pandemic (January 2020 to June 2021), and the predicted volume during the pandemic using 2 prediction models, by operation (primary or revision joint replacement operation) and location (hip or knee).**

| | Primary | | Revision | |
|---|---|---|---|---|
| | Hip | Knee | Hip | Knee |
| **Pre-COVID** | | | | |
| 2016–2019 | 2,623 | 2,674 | 413 | 334 |
| 2018–2019 | 1,205 | 1,183 | 189 | 172 |
| **COVID-19** | | | | |
| Observed | 443 | 350 | 69 | 58 |
| Prediction 1 –TSLM 2016–2019 | 758 (58.5%) | 741 (47.2%) | 90 (76.6%) | 139 (41.7%) |
| Prediction 2 –TSLM 2018–2019 | 814 (54.4%) | 1,027 (34.1%) | 19 (358%) | 156 (37.2%) |

TSLM–time series linear model with terms for trend and season

## A comparison of patients treated before and during the COVID-19 pandemic

In our comparison of patients treated between January 2016 and December 2019 (pre-COVID-19) with those treated during the COVID-19 preparation and restriction phases combined, we found a higher proportion of patients having a primary operation during the pandemic had 2 or more comorbidities than those treated before the pandemic (2 comorbidities: 13% versus 10%, 3 or more comorbidities: 4.4% versus 3.0%, P<0.001) (Table 2). The p-value was consistent with very strong evidence against the null hypothesis of no association. A higher proportion of patients having a revision operation during the pandemic had three or more comorbidities than those treated before the pandemic (3 or more comorbidities: 2.7%

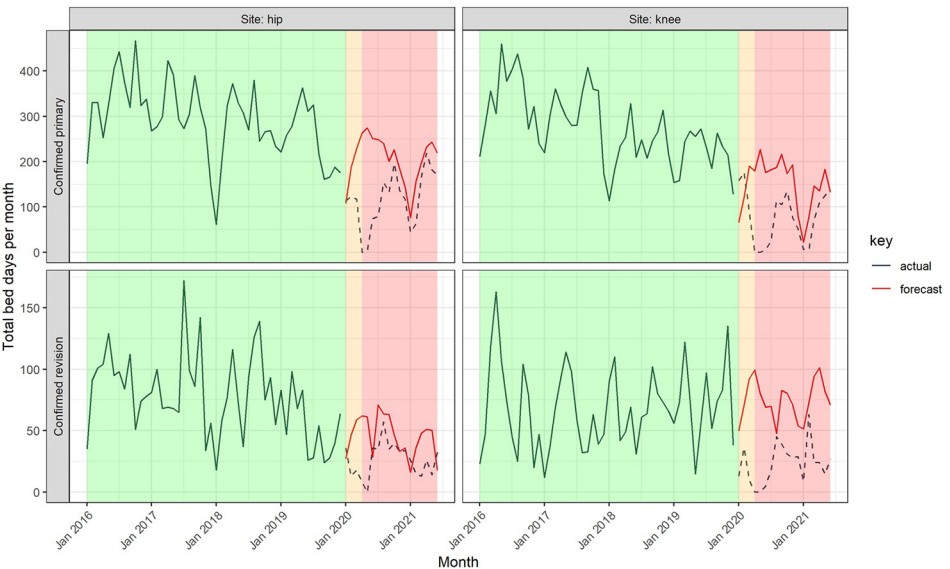

**Fig 3. A comparison of the observed bed occupancy by patients receiving primary and revision elective hip and knee replacements 2016–2019 (solid black) compared with the predicted (solid red) and observed bed occupancy (dashed black) between 1st January 2020 and 30th June 2021 per calendar month, by operation (primary or revision joint replacement operation) and location (hip or knee).** Note–background colours signify phases of the COVID-19 pandemic: green = 'Pre COVID-19', orange = 'COVID-19 preparation', red = 'COVID-19 restrictions'.

**Table 2. A comparison of patients treated by COVID-19 phase and operation type.**

| Characteristic | Confirmed primary | | | | p-value[2] | Confirmed revision | | | | p-value[2] |
|---|---|---|---|---|---|---|---|---|---|---|
| | Pre COVID-19 | COVID-19 | | | | Pre COVID-19 | COVID-19 | | | |
| | | Preparation | Restrictions | Overall | | | Preparation | Restrictions | Overall | |
| | N = 5,297[1] | N = 206[1] | N = 587[1] | N = 793[1] | | N = 747[1] | N = 24[1] | N = 103[1] | N = 127[1] | |
| **Sex** | | | | | 0.473 | | | | | 0.848 |
| Female | 3,197 (60%) | 107 (52%) | 361 (61%) | 468 (59%) | | 399 (53%) | 13 (54%) | 56 (54%) | 69 (54%) | |
| Male | 2,100 (40%) | 99 (48%) | 226 (39%) | 325 (41%) | | 348 (47%) | 11 (46%) | 47 (46%) | 58 (46%) | |
| **Age on Admission** | 69 (59, 76) | 66 (59, 74) | 69 (57, 76) | 68 (58, 76) | 0.082 | 72 (63, 78) | 66 (61, 74) | 69 (60, 77) | 68 (60, 76) | 0.017 |
| **Number of CCI conditions** | | | | | <0.001 | | | | | 0.088 |
| 0 | 2,923 (55%) | 100 (49%) | 275 (47%) | 375 (47%) | | 396 (53%) | 15 (62%) | 50 (49%) | 65 (51%) | |
| 1 | 1,674 (32%) | 73 (35%) | 209 (36%) | 282 (36%) | | 250 (33%) | 5 (21%) | 38 (37%) | 43 (34%) | |
| 2 | 539 (10%) | 27 (13%) | 74 (13%) | 101 (13%) | | 81 (11%) | 1 (4.2%) | 9 (8.7%) | 10 (7.9%) | |
| 3 or more | 161 (3.0%) | 6 (2.9%) | 29 (4.9%) | 35 (4.4%) | | 20 (2.7%) | 3 (12%) | 6 (5.8%) | 9 (7.1%) | |
| **IMD** | | | | | 0.720 | | | | | 0.076 |
| Least deprived | 1,424 (27%) | 61 (30%) | 170 (30%) | 231 (30%) | | 196 (27%) | 9 (38%) | 35 (35%) | 44 (35%) | |
| Less | 1,248 (24%) | 52 (25%) | 131 (23%) | 183 (24%) | | 188 (26%) | 4 (17%) | 19 (19%) | 23 (18%) | |
| Middle | 889 (17%) | 29 (14%) | 100 (17%) | 129 (17%) | | 145 (20%) | 3 (12%) | 23 (23%) | 26 (21%) | |
| More | 897 (17%) | 38 (19%) | 91 (16%) | 129 (17%) | | 125 (17%) | 3 (12%) | 11 (11%) | 14 (11%) | |
| Most deprived | 759 (15%) | 25 (12%) | 81 (14%) | 106 (14%) | | 80 (11%) | 5 (21%) | 13 (13%) | 18 (14%) | |
| Unknown | 80 | 1 | 14 | 15 | | 13 | 0 | 2 | 2 | |
| **Surgery site** | | | | | | | | | | |
| Site: hip | 2,623 (50%) | 90 (44%) | 353 (60%) | 443 (56%) | | 413 (55%) | 10 (42%) | 59 (57%) | 69 (54%) | |
| Site: knee | 2,674 (50%) | 116 (56%) | 234 (40%) | 350 (44%) | | 334 (45%) | 14 (58%) | 44 (43%) | 58 (46%) | |
| **LOS (days)** | 4 (3, 6) | 3 (2, 5) | 3 (3, 5) | 3 (3, 5) | <0.001[2] | 6 (4, 12) | 5 (2, 7) | 5 (3, 9) | 5 (3, 9) | 0.006[2] |
| | - | | | 0.85 (0.82, 0.88)[3] | <0.001[3] | - | | | 0.80 (0.75, 0.86)[3] | <0.001[3] |

IMD: Indices of Multiple Deprivation, CCI: Charlson Comorbidity Index, LOS: Length of stay

[1]n (%); Median (IQR)

[2]P-values from Pearson's Chi-squared test, Wilcoxon rank sum test and Fishers exact test comparing patients treated during the 'Pre COVID-19' phase with those treated during the combined COVID-19 preparation and restriction phases

[3]Incidence risk ratios, 95% confidence intervals and P-values from Poisson regression models adjusted for age, gender, IMD and number of comorbidities

Pre COVID-19 versus 7.1% COVID-19), but the trend was not consistent for fewer comorbidities (No comorbidities: 53% Pre COVID-19 versus 51% COVID-19, P = 0.088). Revision operations were performed on younger patients during the pandemic than before (68 versus 72 years, P = 0.017), and the p-value was consistent with strong evidence against the null hypothesis. There were no differences in the sex distribution of patients treated during the pandemic compared with before the pandemic. A higher proportion of revisions were performed on people from the least deprived areas during the pandemic compared with before the pandemic (35% versus 27%) but otherwise differences with respect to IMD were inconsistent, with a large p-value for the comparison of primaries (0.720) and a p-value for the comparison of revisions (0.076) consistent with weak evidence against the null hypothesis.

Median length of hospital stay was shorter during the pandemic than before for both primary (three versus four days, P<0.001, Table 1 and Fig 4) and revision operations (five versus six days, P = 0.006). The adjusted Poisson regression models confirmed a reduced risk of longer hospital stays for patients treated during the pandemic compared with those before (incidence risk ratio [IRR] = 0.84; 95% CI: 0.81, 0.87; P<0.001 for primaries; IRR = 0.81;95% CI:

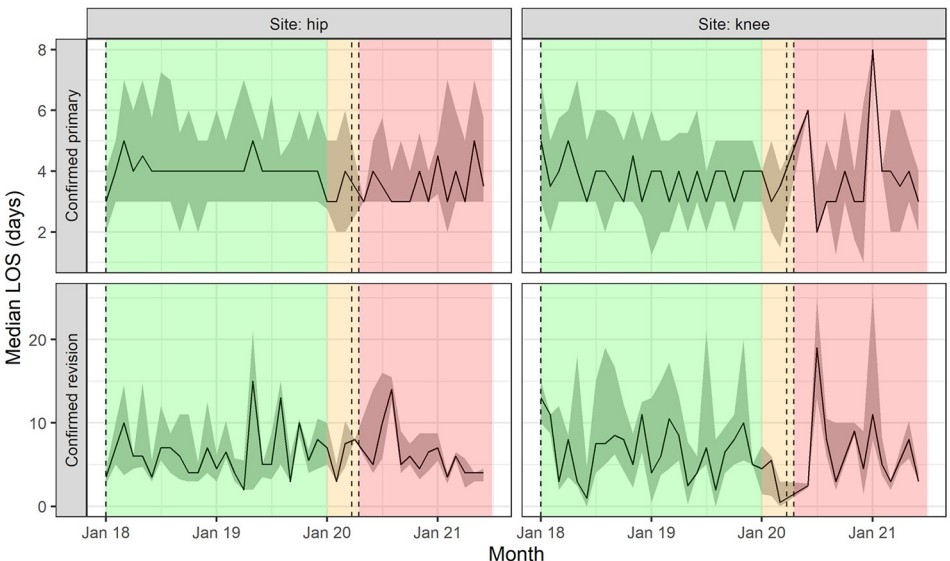

**Fig 4. Median length of stay by operation type and site of surgery.** Note–background colours signify phases of the COVID-19 pandemic: green = 'Pre COVID-19', orange = 'COVID-19 preparation', red = 'COVID-19 restrictions'; shading signifies the span from the lower (25%) to upper (75%) quartiles.

0.75, 0.86; P<0.001 for revisions). Comparisons using a shorter pre-COVID 19 time period (from 01/01/2018 onwards) showed similar results (S5 Table).

A higher proportion of patients receiving primary or revision procedures during the pandemic had three or more comorbidities compared with before (primary: 3.7% versus 4.4%, revision: 3.3% versus 7.1% for Pre-COVID-19 and COID-19 respectively), but there was no clear pattern between the proportion of patient with up to two comorbidities and COVID-19 phase. The associated p-values were consequently large, providing no evidence against the null hypothesis. Post-hoc pairwise comparisons of the comorbidities of patients having primary joint replacements using chi-squared tests, corrected with the Benjamini and Hochberg method, indicated that for primaries no comorbidities versus any comorbidities were statistically significantly different (p-values<0.05, results not reported) whereas there was no difference between the groups with one, two or 3+ comorbidities. For revisions there was weak evidence of differences between three or more comorbidities compared with none, one or two comorbidities (p-values = 0.065, results not reported), with no evidence of differences between groups with two or fewer comorbidities.

## Discussion

Patients who had an elective primary or revision hip or knee replacement between January 2020 and June 2021 at one large teaching hospital were younger than those treated before the COVID-19 pandemic, had more comorbidities, but also stayed in hospital for less time. We found no indication that the prioritisation of patients for surgery during the pandemic has introduced new inequalities in access to elective hip and knee replacements with respect to area-level deprivation. We found that the volume of elective hip and knee primary and revision replacements between January 2020 and June 2021 was 53% of the expected volume.

The reduction in volume of primary and revision hip and knee replacements at our study hospital is similar to the impact across England and Wales in 2020 compared with 2019 (56% of expected hips and 48% of expected knees) [29], and in other countries [30–32]. We have

further highlighted a potentially concerning impact on revision operations as was observed in Poland [32], although we observed a greater impact on revisions of primary knee rather than hip replacements. Thaler and colleagues suggested that the reduced volume of revision procedures may be due to the lack of availability of intensive care units [33]. Although our predictions of expected revisions were more challenging than predictions of expected primaries, and findings may be less reliable. Primary joint replacements may need to be revised for different reasons, including for aseptic loosening, pain, dislocation, infection, and periprosthetic fracture [2]. Delays to revision operations longer than six months are likely to cause significant increases in pain and physical disability [34]. Revision operations are more complex than primary joint replacements and need a longer in-hospital recovery period [10]. Although these operations are much lower in volume, the reduced volume of revision operations may be of particular concern to Trusts if the trend we observed is replicated elsewhere.

We found no change in the area-level deprivation of patients treated during the pandemic compared with those treated before. This contrasts with a study at the same Trust of the impact of the suspension of elective operations during the 2017/2018 flu season, which found a reduction in the proportion of people from the most deprived areas having knee replacements [35]. To our knowledge there are no other studies of the impact of suspending hip and knee replacements on inequality in access to surgery. Patients treated during the pandemic were younger than those treated before the pandemic, in contrast to studies from other countries which found either a mixed effect of age depending on the joint being replaced [31] or no difference in age [36]. We also found that patients treated during the pandemic had more comorbidities. Younger patients in Spain who had their surgery re-scheduled in early 2020 were more willing to undergo surgery during the pandemic than older patients [37]. However, this did not extend to patients with comorbidities considered high risk with respect to possible complications from a COVID-19 infection. Further research regarding the epidemiology of hip and knee replacements during the pandemic may confirm whether our findings have been observed in other hospital Trusts.

We observed a reduction in the mean length of stay for primary and revision operations of one day during the pandemic compared with before. This is in agreement with findings from the USA [38, 39] and Poland [40] but contrasts with other findings from the UK [41]. This finding was consistent using either four years or two years of pre-pandemic data, suggesting this does not reflect a pre-pandemic trend. This reduction may be due to the selection of patients who were more likely to need a shorter length of stay, reflected in the younger patients treated during the pandemic. But the presence of comorbidities, which was higher for those treated during the pandemic, is associated with longer hospital stays [42]. We adjusted for patient factors, including number of comorbidities, which did not alter this finding. The shortened length of stay is unlikely to have been caused by pressure for beds from other hospital specialties but could be a consequence of infection control measures intended to minimise in-hospital transmission of COVID-19 infections. We do not know whether these patients had more post-operative complications or whether their recovery was adversely affected by their earlier discharge. If not, the safe introduction of shorter hospital stays could be a valuable finding to support the challenge of addressing the waiting list for elective hip and knee replacements.

## Strengths and limitations

Analyses of time-series using data from electronic health records of hospital Trusts are an informative way of illustrating to health service managers and clinicians how service provision has been impacted by events such as the COVID-19 pandemic. However, we only have data

from one NHS Trust and our findings may not generalise to other NHS Trusts. We do not have any information about activity in the independent sector and the extent to which primary operations have been outsourced from the NHS. We do not know how the waiting list has been affected during the pandemic or the details of the patients waiting for their surgery. This was outside the scope and permissions for this study but will be an important consideration for local and national planning for the post-pandemic recovery of elective surgery provision. Fewer revision than primary operations are performed at the Trust and therefore the reliability of predictions from historic activity may be less reliable, as was seen in the variation in predicted volume of revision hip replacements using 48 or 24 months of preceding activity. We did not model temporal trends in demographic and clinical characteristics during the course of the pandemic, instead we assumed no temporal change within each phase of the pandemic. We have focussed on elective hip and knee replacements, but the effects of restrictions on elective operations will clearly be felt more widely. Finally, we have not compared patient outcomes for those treated during the pandemic with those treated earlier. Surgery is likely to have been delayed for those who had their operation during the pandemic, and their experiences and outcomes may give an insight into what the outcomes might be for those still on the waiting list.

## Conclusions

Despite the significant impact of the pandemic on the provision of primary and elective hip and knee replacements, we found no evidence of new inequalities in access to these operations with respect to area-level deprivation. The shorter length of stay observed during the pandemic may inform changes to the postoperative management of patients and influence plans to address the waiting list for these elective operations. Further research looking at the impact of the pandemic on other elective operations and on the safety of shorter lengths of stay implemented during the pandemic may be valuable.

## Supporting information

**S1 Table. OPCS-4 codes used to identify primary hip and knee replacement operations.**
(DOCX)

**S2 Table. A list of ICD10 codes used to identify conditions in the CCI.**
(DOCX)

**S3 Table. A description of the study sample by COVID-19 phase.**
(DOCX)

**S4 Table. Proportion patients with each of the conditions included in the CCI, by joint and type of surgery.**
(DOCX)

**S5 Table. A description of the study sample by COVID-19 phase and operation type.** Pre COVID-19 phase restricted to operations performed on or after 01/01/2018.
(DOCX)

## Acknowledgments

The authors are grateful to the members of the public who contributed through the patient and public involvement process and for Mike Bell and Carmel McGrath for organising the workshop.

## Author Contributions

**Conceptualization:** Chris M. Penfold, Ashley W. Blom, Maria Theresa Redaniel, Andrew Judge.

**Data curation:** Tim Keen, Andrew Elliott.

**Formal analysis:** Chris M. Penfold.

**Funding acquisition:** Ashley W. Blom, Andrew Judge.

**Methodology:** Chris M. Penfold, Tim Jones, Emily Eyles, Andrew Judge.

**Writing – original draft:** Chris M. Penfold.

**Writing – review & editing:** Chris M. Penfold, Ashley W. Blom, Maria Theresa Redaniel, Tim Jones, Emily Eyles, Tim Keen, Andrew Elliott, Andrew Judge.

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
