## [Decision Letter · Decision Letter 0]

5 Dec 2022

PONE-D-22-21190The impact of restricted provision of elective operations during the COVID-19 pandemic on patient selection for elective NHS hip and knee replacementsPLOS ONE

Dear Dr. Penfold,

Thank you for submitting your manuscript to PLOS ONE. After careful consideration, we feel that it has merit but does not fully meet PLOS ONE’s publication criteria as it currently stands. Therefore, we invite you to submit a revised version of the manuscript that addresses the points raised during the review process.

ACADEMIC EDITOR: Please address well to the comments.

We look forward to receiving your revised manuscript.

Kind regards,

Ka Chun Chong

Academic Editor

PLOS ONE

https://journals.plos.org/plosone/s/fileid=ba62/PLOSOne_formatting_sample_title_authors_affiliations.pdf.

“This study was funded by the HDRUK Better Care Partnership (#6.12). This research was supported by the NIHR Biomedical Research Centre at University Hospitals Bristol and Weston NHS Foundation Trust and the University of Bristol (CP, AJ, AB) and by the National Institute for Health Research (NIHR) Applied Research Collaboration West (NIHR ARC West, MTR, TJ, EE). The views expressed in this article are those of the author(s) and not necessarily those of the NIHR or the Department of Health and Social Care.”

4. Thank you for stating the following in the Funding Section of your manuscript:

“This study was funded by the HDRUK Better Care Partnership (#6.12). This research was supported by the NIHR Biomedical Research Centre at University Hospitals Bristol and Weston NHS Foundation Trust and the University of Bristol and by the National Institute for Health Research (NIHR) Applied Research Collaboration West (NIHR ARC West). The views expressed in this article are those of the author(s) and not necessarily those of the NIHR or the Department of Health and Social Care.”

“This study was funded by the HDRUK Better Care Partnership (#6.12). This research was supported by the NIHR Biomedical Research Centre at University Hospitals Bristol and Weston NHS Foundation Trust and the University of Bristol (CP, AJ, AB) and by the National Institute for Health Research (NIHR) Applied Research Collaboration West (NIHR ARC West, MTR, TJ, EE). The views expressed in this article are those of the author(s) and not necessarily those of the NIHR or the Department of Health and Social Care.”

“I have read the journal’s policy and the authors of this manuscript have the following competing interests: AB (Stryker) has received research and other financial support from companies or suppliers outside the submitted work. AJ declares advisory board positions with receipt of fees (Anthera Pharmaceuticals, INC.) and paid consultancy work (Freshfields Bruckhaus Deringer) for companies outside the submitted work. All other authors declare no competing interests. This does not alter our adherence to PLOS ONE policies on sharing data and materials.”

7. We note that you have indicated that data from this study are available upon request. PLOS only allows data to be available upon request if there are legal or ethical restrictions on sharing data publicly. For more information on unacceptable data access restrictions, please see http://journals.plos.org/plosone/s/data-availability#loc-unacceptable-data-access-restrictions.

8. Your ethics statement should only appear in the Methods section of your manuscript. If your ethics statement is written in any section besides the Methods, please delete it from any other section.

Reviewers' comments:

Reviewer's Responses to Questions

**Comments to the Author**

1. Is the manuscript technically sound, and do the data support the conclusions?

Reviewer #1: Yes

Reviewer #2: Yes

Reviewer #3: Yes

Reviewer #4: Yes

2. Has the statistical analysis been performed appropriately and rigorously? 

Reviewer #1: I Don't Know

Reviewer #2: Yes

Reviewer #3: Yes

Reviewer #4: Yes

3. Have the authors made all data underlying the findings in their manuscript fully available?

Reviewer #1: No

Reviewer #2: Yes

Reviewer #3: Yes

Reviewer #4: No

4. Is the manuscript presented in an intelligible fashion and written in standard English?

Reviewer #1: Yes

Reviewer #2: Yes

Reviewer #3: Yes

Reviewer #4: No

5. Review Comments to the Author

Reviewer #1: Thank you for the opportunity to review this manuscript that provide information about the impact of the COVID-19 in the hospital care for hip and knee replacements in NHS. This study found there were significant reduction in the volume of the operations during the COVID-19, differences in patients characteristics and time the patients stay in the hospital. I have several comments and questions for their consideration.

Title

What is the NHS in the title? Consider to use the full name.

Abstract

HR -NR in abstract. I would suggest to use full name and no abbreviations.

In methods the author put 2016/01- 2021/06 what the author means?

What the author means with 2016 to 2021?

Introduction

90 what that is mean? (line3)

NHS? what that it mean?

the author uses a lot of symbols, I would consider writing the meaning of the symbols apart of using “ >” ,“~”

what the literature says about age when this surgery was performed?

Methods

The authors should describe better the teaching hospital, location, level of care...

Specify better the “Covid restrictions” period.

What are the Knee and hip replacements indications?

What are the difference between primary surgery and revision surgery?

Results

Uniform the terminology of the surgery sometimes you use hip and knee replacement then you change to hip and knee surgeries.

I would like to know which comorbidities where more common, or type of the comorbidities or examples. you only say 2 or 3 comorbidities but do not specify?

Others

How the selection of patients to surgery was made?

Did covid test or covid influence in the access to surgery?

Where the hospitals had others restrictions or limitations for the patients to come to hospitals or they may go for others hospitals…., was the number of surgeries decrease duo to cancelations? No clinical staff??? The reduction of stay in the hospital was 1 of the hospital policies?

Reviewer #2: In general, the subject of the manuscript is very good besides the authors used the suitable statistical analysis but I have some comments

1.The authors mentioned that since numbers in some categories were very low. They compared characteristics using the Wilcoxon rank sum test for continuous variables (age and length of hospital stay). This is not an accurate paragraph. The Wilcoxon rank sum test should be used when the data are not normally distributed. Otherwise, the independent t-test is the preferred test.

2.The means of age and length of hospital need to adding the standard deviation

3.The paper included a p-value as one number after the dot and I think adding two numbers is more suitable.

4- I think it is preferred for the authors to state that they used the Chi-square test of independence to determine whether there is an association between categorical variables (i.e., whether the variables are independent or related).

5. It is better for authors to include the equations of time-series linear regression models in the paper in order not to waste their efforts

Reviewer #3: Thank you for inviting me to review this manuscript; I have the followings Comments:

Introduction: the statement “Seasonal spikes in hospital admissions…” Explain and support it with data”

The objectives of the study should be rewritten to be more specific and relevant to the outcome of the study rather than describe.

Methods:

How were the data extracted?

What was the sampling technique?

The statement under setting “ (hip: n~240/year; knee: n~270/year) and 25-40% of the total number performed in city” Which city?

Explain CCI, IMD and comment on their validity and these must be supported with references.

The statement “Since we did not model the temporal trend in demographic and clinical characteristics, we were assuming that there was no change in these characteristics within each of the phases of the pandemic” It should be moved under limitations of the study. Furthermore, assumption was inappropriate to mention in this study, please revise.

Tables: Explain in the text the demography of the participants shown in Table 1 .

What was IMD stand for in Table 1?. This was mentioned in the text, and it should be also inserted under Table 1.

What were CCI and LOS stand for in Table 2?. These were mentioned in the text and it should be also inserted under Table 2. Also, In the text, give examples of comorbidities mentioned in Table 2.

Explain the main findings of figures 1, 2 and 3.

Explain the findings of p-values, for example P=0.09, P=0. 017..etc.

The statement in the last lines of results “The trend towards patients with more comorbidities being treated during the pandemic persisted for primary but not revision operations.” Was this investigated in this study or was it an assumption”

Discussion:

It should be revised, and authors should mention each finding rather than listing all findings once and then contrast each with relevant published studies. For example authors quoted “….elective spinal surgery….” under reference 28 which was irrelevant to knee and or hip surgery.

Under discussion, the statement “We observed a reduction in the mean length of stay for primary and revision operations of one day during the pandemic compared with before.” Was this investigated in this study as it was not mentioned under the results or Tables of this study.

Strengths and Limitations:

Start with the strengths of your study as nothing in this regard was mentioned.

Conclusions:

The statement “……. we found no evidence of new inequalities in access to these operations” Was this investigated in the study? If yes it should be highlight and explained under the results of this study.

General comments:

Ethics statement should be explained in detail.

This study will benefit from two lines of recommendations.

Reviewer #4: 1. Inclusion and/or exclusion criteria of choosing patients should be added.

2. Results of the abstract need revision. Add most important findings.

3. Add letters to TABLE 2 to show the significant differences within the group.

4. Delete table 1 because table 2 contains the same data in detail.

5. Construct a table to show the predicted volume of elective primary and revision hip and knee replacement operations prior to COVID-19 and compare it to the restriction period.

6. PLOS authors have the option to publish the peer review history of their article (what does this mean?). If published, this will include your full peer review and any attached files.

Reviewer #1: **Yes: **Vanda Amado

Reviewer #2: **Yes: **Firas Rashad Al-Samarai

Reviewer #3: No

Reviewer #4: No

---

## [Author Response · Author response to Decision Letter 0]

20 Oct 2023

Reviewer 1

Thank you for the opportunity to review this manuscript that provide information about the impact of the COVID-19 in the hospital care for hip and knee replacements in NHS. This study found there were significant reduction in the volume of the operations during the COVID-19, differences in patients characteristics and time the patients stay in the hospital. I have several comments and questions for their consideration.

We thank the reviewer for their careful review of our paper.

Title

What is the NHS in the title? Consider to use the full name.

Page 1, lines 1-2: We have replaced NHS with ‘publicly funded’

Abstract

HR -NR in abstract. I would suggest to use full name and no abbreviations.

Page 3: We have replaced HR and KR with their full equivalents, hip replacement and knee replacement.

In methods the author put 2016/01- 2021/06 what the author means?

Page 3: We have replaced the shortened dates with full year and month.

What the author means with 2016 to 2021?

We hope our response to the comment above provides clarity on this point as well.

Introduction

90 what that is mean? (line3)

Page 5, line 91: We have updated this text as follows: ‘They are effective, with approximately 90% of patients…’

NHS? what that it mean?

Page 5, line 102: We have updated this text as follows: ‘Seasonal spikes in hospital admissions have been a priority challenge for the National Health Service (NHS) for a number of years’

the author uses a lot of symbols, I would consider writing the meaning of the symbols apart of using “ >” ,“~”

We have removed most symbols used in text and tables, which we hope has improved the readability.

what the literature says about age when this surgery was performed?

Page 5, lines 93-99: We have included the typical age and gender split of primary hip and knee replacement patients.

Methods

The authors should describe better the teaching hospital, location, level of care...

Page 6, lines 130: We have provided more detail about the location of the hospital and its size.

Specify better the “Covid restrictions” period.

Page 6, lines 147-152: We have included more detail about the national guidance for patients undergoing elective procedures requiring anaesthesia which applied to the Covid restrictions period.

What are the Knee and hip replacements indications?

Page 5, line 90: Osteoarthritis is the main or only indication in around 90% hip replacements and 95% knee replacements. We have amended the introduction as follows: “Elective hip and knee replacements are two of the most commonly performed operations in the UK to relieve the pain and functional limitations of OA (an indication for surgery in more than 90% cases) [2–4]”

What are the difference between primary surgery and revision surgery?

Page 5, lines 95-97: We have included a more detailed explanation of primary and revision surgery in the introduction.

Results

Uniform the terminology of the surgery sometimes you use hip and knee replacement then you change to hip and knee surgeries.

We have standardized the terminology throughout.

I would like to know which comorbidities where more common, or type of the comorbidities or examples. you only say 2 or 3 comorbidities but do not specify?

We have included a table (Supplementary Table T4) in the supplementary material detailing the proportion of hip and knee primary and revision patients with each of the CCI comorbidities, and have summarized this in the cohort description.

Others

How the selection of patients to surgery was made?

We are unsure specifically what the reviewer is asking. If this relates to the indications for surgery, we hope our inclusion of more detail about the indications for surgery in the introduction is sufficient.

Did covid test or covid influence in the access to surgery?

Page 6, lines 148-152: We have included the details of NICE guidelines introduced in June 2020, including the requirement that patients have a SARS-CoV-2 test 3 days before admission.

Where the hospitals had others restrictions or limitations for the patients to come to hospitals or they may go for others hospitals…., was the number of surgeries decrease duo to cancelations? No clinical staff??? The reduction of stay in the hospital was 1 of the hospital policies?

We thank the reviewer for this comment. We agree that the reduction in number of operations performed may have been affected by hospital organizational factors or staff sickness, among other factors. However, our study was not intended to explore the specific reason for reduced surgical volume, but to describe how this changed during the pandemic and the impact on which patients had surgery. Organizational factors which may better explain why surgical volume was reduced was beyond the scope and permissions of our study.

Reviewer 2

Dear Sir,

In general, the subject of the manuscript is very good besides the authors used the suitable statistical analysis but I have some comments

1.The authors mentioned that since numbers in some categories were very low. They compared characteristics using the Wilcoxon rank sum test for continuous variables (age and length of hospital stay). This is not an accurate paragraph. The Wilcoxon rank sum test should be used when the data are not normally distributed. Otherwise, the independent t-test is the preferred test. 

Thank you for your feedback and comments.

Page 8, lines 196-198: Our wording of this part of the methods has caused a mis-interpretation. We have therefore changed the wording as follows: “We compared the demographics and clinical characteristics of patients who received operations during the pre-COVID 19 period with those treated either in the COVID-19 preparation or COVID-19 restrictions periods. Since numbers in some categories were very low we combined the COVID-19 preparation and COVID-19 restrictions periods. We compared characteristics using the Wilcoxon rank…”

2.The means of age and length of hospital need to adding the standard deviation

We have included a footnote to Table 1 (now Supplementary Table T3) indicating the age and length of hospital stay are reported as median (25%, 75%)

3.The paper included a p-value as one number after the dot and I think adding two numbers is more suitable.

We have changed all p-values to include 3 decimal places.

4- I think it is preferred for the authors to state that they used the Chi-square test of independence to determine whether there is an association between categorical variables (i.e., whether the variables are independent or related).

We thank the reviewer for this comment. We have made changes to this paragraph (‘Comparison of patient demographics and clinical characteristics’) in response to other comments which we hope have aided in the clarity of the statistical analyses we performed.

5. It is better for authors to include the equations of time-series linear regression models in the paper in order not to waste their efforts

Thank you for this comment. Unfortunately we are unclear what clarification the reviewer is asking for here. To our knowledge, the explanation of the time series linear model as stated is appropriate.

Reviewer 3

Manuscript Number: PONE-D-22-21190

Re “The impact of restricted provision of elective operations during the COVID-19 pandemic on patient selection for elective NHS hip and knee replacements”

Thank you for inviting me to review this manuscript; I have the followings Comments:

Thank you to the reviewer for their comments.

1. Introduction: the statement “Seasonal spikes in hospital admissions…” Explain and support it with data” 

Page 5, lines 102-109: We have included a further illustration of seasonal spikes in bed occupancy during winter, in addition to the included examples of impacts on A&E departments, emergency funding provision and NHS management plans.

2. The objectives of the study should be rewritten to be more specific and relevant to the outcome of the study rather than describe.

Page 6, lines 122-126: We have revised the objectives to be more specific and relevant to the outcome of the study.

Methods: 

1. How were the data extracted?

Page 6, lines 139-141: Data were from an extract of elective primary hip and knee replacement inpatient admissions identified from the Trust’s electronic medical records. We have amended the description of the ‘Study sample’ accordingly.

2. What was the sampling technique? 

We did not sample from the population, but rather included all hip and knee replacement operations performed in the Trust within the specified timeframe and which met the inclusion and exclusion criteria.

3. The statement under setting “ (hip: n~240/year; knee: n~270/year) and 25-40% of the total number performed in city” Which city?

Page 6, line 130: We have included the name of the City

4. Explain CCI, IMD and comment on their validity and these must be supported with references.

Page 7, lines 169-183: We have included further details of the CCI and IMD, including references

5. The statement “Since we did not model the temporal trend in demographic and clinical characteristics, we were assuming that there was no change in these characteristics within each of the phases of the pandemic” It should be moved under limitations of the study. Furthermore, assumption was inappropriate to mention in this study, please revise.

We mentioned this assumption in our statistical methods in order to explain our decision to repeat our analyses with a shorter pre-COVID 19 period. We have therefore kept this explanation in our methods but have also included it as a potential limitation in our discussion.

6. Tables: Explain in the text the demography of the participants shown in Table 1 . 

Page 9, lines 217-223: We have included a summary of patients treated during the Pre COVID-19 period in the text. Results described later compare patients treated during the ‘COVID-19 restrictions’ period with this group of patients.

7. What was IMD stand for in Table 1?. This was mentioned in the text, and it should be also inserted under Table 1.

Thank you for highlighting this. We have included additional footnotes of the abbreviations used in Table 1 (now Supplementary Table T3) and Table 2 (now Table 1).

8. What were CCI and LOS stand for in Table 2?. These were mentioned in the text and it should be also inserted under Table 2. Also, In the text, give examples of comorbidities mentioned in Table 2.

Please see our response to point 7 above. In addition, we have included examples of the conditions included in the CCI in the text (Page 9, lines 219-221).

9. Explain the main findings of figures 1, 2 and 3.

We have already included a summary of the key findings from these figures in the second paragraph of our results, under the sub-heading “A comparison of observed versus forecast predicted hospital activity” (Page 9, line 225).

10. Explain the findings of p-values, for example P=0.09, P=0. 017..etc. 

We have included further explanations of some of the p-values. However, we have not included an explanation of all since this will add substantially to the text describing the results.

11. The statement in the last lines of results “The trend towards patients with more comorbidities being treated during the pandemic persisted for primary but not revision operations.” Was this investigated in this study or was it an assumption” 

Thank you for this observation. This sentence was incorrectly added to a new paragraph but related to the sensitivity analyses described in the preceding paragraph. Also, following a misreporting of the count of comorbidities, the results of the sensitivity analyses have changed and we have updated the interpretation accordingly.

12. Discussion:

a. It should be revised, and authors should mention each finding rather than listing all findings once and then contrast each with relevant published studies. For example authors quoted “….elective spinal surgery….” under reference 28 which was irrelevant to knee and or hip surgery.

Page 11, line 299 to page 12, line 342: We have rearranged the discussion in the same order in which we presented the results. We have also increased comparison with published literature. We have also removed the comparison with elective spinal surgery.

b. Under discussion, the statement “We observed a reduction in the mean length of stay for primary and revision operations of one day during the pandemic compared with before.” Was this investigated in this study as it was not mentioned under the results or Tables of this study.

Page 10, lines 271-277: The reduced length of stay was described in the results in the paragraph beginning “Median length of hospital stay was shorter during the pandemic than before for both primary (three versus four days, P<0.001, Table 1 & Figure 4) and revision operations (five versus six days, P=0.006).”.

13. Strengths and Limitations:

a. Start with the strengths of your study as nothing in this regard was mentioned.

Page 12, lines 343-346: The Strengths and Limitations paragraph has been updated.

14. Conclusions:

a. The statement “……. we found no evidence of new inequalities in access to these operations” Was this investigated in the study? If yes it should be highlight and explained under the results of this study.

Page 13, lines 365-366: We have amended this sentence to be clear this relates to area-level deprivation, as follows:

“we found no evidence of new inequalities in access to these operations with respect to area-level deprivation”

15. General comments: 

a. Ethics statement should be explained in detail.

Page 6, lines 134-137: We have included more detail about the ethical approvals.

b. This study will benefit from two lines of recommendations.

Page 13, lines 364-370: We have reorganised the Conclusions to highlight two recommendations for further research as follows:

“Further research looking at the impact of the COVID-19 pandemic on other elective operations and on the safety of shorter lengths of stay implemented during the pandemic may be valuable.”

Reviewer 4

We thank reviewer 4 for their helpful comments.

1. The title is too long, I suggest to change it to:

a. The impact of restricted provision of elective hip and knee joints replacement during pandemic of COVID-19 in England.

Thank you for this comment. We have changed the title as per your suggestion.

2. Please revise this sentence to:

a. We included all elective primary and revision HRs and KRs performed at one hospital between January 2016 to June 2021.

Page 3, lines 72-73: We have changed this sentence as suggested.

3. Change it to: for the years 2016-2019,

Page 3, line 64: Done

4. Please revise to: for the period April 2020 to June 2021

Page 3, line 65: I have updated the text to “January 2020 to June 2021” since this is the period over which we predicted the expected surgical activity.

5. April 2020 to June 2021

Page 3, line 73: I have updated the text to “January 2020 to June 2021” since this is the period over which we predicted the expected surgical activity.

6. Revise to: elective hip and knee replacements.

Page 3, lines 67-68: Done

7. previously operated patients

Page 3, lines 68-70: We have clarified this sentence with the inclusion of ‘…the pandemic…’

8. This number 6964 of operations for whole period 2016-2021 (included patients during pandemic) or for the period 2016-2019? Please clarify in the text.

Page 3, lines 72-73: We have clarified in the text that this is the total number of operations performed between January 2016 and June 2021.

9. I don't see any comparative between predicted data and the data during pandemic!!!

Page 3, lines 73-75: In our ‘Results’ paragraph we included the sentence ‘Primary HR volume was 59% of predicted, and 47% for primary KRs. Revision HR volume was 77% of predicted, and 42% for revision KR’. This incorporates the comparison of observed to predicted volume of operations during the pandemic.

10. before the pandemic.

Page 3, line 78: Done

11. It is repeated findings, please add main conclusion in your study.

Page 3, lines 80-81: We have included a main conclusion.

12. You need to show that in the results....See previous comment.

We hope our clarification provided in response to comment 9 above reassures the reviewer that these results are reported in the abstract.

13. Remove the hyphen.

Page 5, line 88: Done

14. Change the parentheses to square brackets throughout the manuscript.

Done

15. Explain the difference between primary and revision H/KR after the sentence.

Page 5, lines 95-97: Done

16. How about the revision? How many days? Add reference.

Page 5, lines 99-101: We have amended the opening paragraph as follows: “The mean length of hospital stay after primary hip and knee replacements is approximately three days [9], and is approximately 1-1.5 days longer for revision operations [10].”

17. Revise to: National Health Service (NHS)

Page 5, line 104: Done

18. Revise to: 2016/2017

Page 5, line 105: Done

19. Revise to: Accidents and Emergencies

Page 5, line 105: Done

20. 2012

Page 5, line 106: Done

21. Revise to: 2018, more than

Page 5, lines 106-107: Done

22. Is this during pandemic or before (which years?)?

Page 5, lines 108-109: We have amended this sentence as follows: “Management plans have been extended before the COVID-19 pandemic to include increasing bed occupancy above operating limits (85%) and cancelling elective operations [14].”

23. ,

Page 5, line 110: Done

24. 2018

Page 5, line 111: Done

25. or between January to March?

Page 5, line 112: To our knowledge the advice from NHS England applied only to January 2018.

26. Please remove this, these are not research questions, it is just repeating the aim of the study.

Page 6, lines 122-126: In response to Reviewer 3 we have re-written these objectives to be more specific and relevant.

27. How about the revision operations? What is the ratio of primary/revision?

Page 6, lines 130-133: Unfortunately the publicly available data for the National Joint Registry we used for these figures only reports the previous three years of activity, which has been updated and now does not cover the pre-pandemic period. We are therefore unable to add further detail on revision operations and the ratio of primary to revision operations.

28. Which city? Please clarify......

Page 6, line 130: Done

29. Please state the inclusion and/or exclusion criteria for selecting patients.

We have already included the full list of OPCS4 procedure and surgical site codes used to identify primary and revision hip and knee replacement procedures eligible to be included, and have specified that non-elective admissions and records which were not unambiguous with respect to site of surgery or type of operation would be excluded. No further inclusion or exclusion criteria were used for selecting patients.

30. This sentence repeated many times. Please remove this.

Page 6, lines 147: We have removed this sentence and incorporated the key points into the following sentence.

31. Revise to: the period from 15th April 2020.

Page 6, line 147: Done

32. Remove

Page 6, line 148: Done

33. What was the purpose of this period in your study, please explain in the text.

Page 7, lines 154-156: We have amended this sentence as follows:

“We have therefore designated the period from 1st January to 15th April 2020 as “COVID-19 preparation”, in recognition of the reorganisation of the provision of hospital treatments and preparation for the introduction of COVID-19 restrictions...”

34. This period included 4 years, however, the COVID-19 restriction period included only 14 months in your study. According to your results (Figure 1), there was a fluctuation in the number of operations. So, it was more practical to use similar period of COVID-19 restriction in Pre-COVID-19 period (also 14 months) rather than 4 years to predict accurate data and the comparison is more accurate.

We agree with the reviewer that the length of time used to model the Pre-COVID-19 period will affect the predicted volume of operations expected during the pandemic, particularly in the presence of apparent hospital-level changes in the volume of operations or types of patients treated. 

We respectfully disagree with the reviewer that 14 months would be an appropriate Pre-COVID-19 period, since this would include only one full season and may limit our modelling of seasonal effects. We have included an additional sensitivity analysis in which we use the 24 months of surgical activity to predict the expected volume of surgery during COVID. We have included these results in our new table (Table 1) and amended the relevant text in the Methods (Page 8, lines 203-208) and Results (Page 9, lines 244-252). We have also included a further limitation in our Discussion relating to the instability of predictions of expected hip revisions, and the subsequent estimate of observed reduction in activity.

35. This period also used for prediction the number of operations. Please revise the sentence.

We feel the reviewer may have been referring to the “COVID-19 preparation” period. We have amended our description of this period above, which we hope clarifies the purpose of this period. The following section of the Methods also details that this was combined with the “COVID-19 restriction” period for the purposes of calculating the observed activity.

36. Please revise to: 

a. We calculated the number of pre-existing conditions (comorbidities) in patients from those included in the Charlson Comorbidity Index (CCI).

Page 7, lines 171-174: We amended this sentence to ‘We counted the number of pre-existing conditions (comorbidities) in patients from those included in the Charlson Comorbidity Index (CCI)’

37. Analysis

Page 8, line 184: Done

38. Why did you use the non-parametric test and not the parametric one?

Page 8, lines 197-198: We have clarified that these tests were used for non-normal continuous variables.

39. This sentence is also repeated many times. Please remove.

Page 9, lines 216-217: We have respectfully chosen to keep this sentence in order to summarise in the text the total number of operations included in our study and the time period over which these were performed. 

40. All these mentioned in the table1 and 2. Please remove this paragraph and focus on what is important in your results.

Page 9, line 219: We have removed these results. Note we have also included additional detail about the cohort as requested by Reviewer 3.

41. Add comma ,

Page 9, line 228: Done

42. Please construct a table to show the predicted volume of elective primary and revision hip and knee replacement operations prior COVID-19 and compare it with the restriction period.

Table 1: We have included an additional table comparing the predicted number of operations using different pre-COVID-19 time periods.

43. Change forecast to: predicted. Don't use two terms for the same meaning.

Done

44. All these results present in the table 2, so please remove this paragraph. Also, I suggest to separate table 2 to two parts, one of them includes number of comorbidities, sex, and age.

Page 9, line 226 onwards: Thank you for this comment. This paragraph contains the pertinent results relating to the age and number of comorbidities for patients having primaries and revisions. These results form the basis of some of the discussion points. We have therefore chosen to keep these results in the text for the benefit of readers and to keep this as a single table.

45. Please indicate the important data related to IMD, at least refer to the table describing this topic.

Page 10, lines 266-270: We have expanded our summary of the important results relating to IMD.

46. Please arrange the discussion paragraphs in the same order in which you presented your results.

We have rearranged the discussion in the same order in which we presented the results. We have also increased comparison with published literature.

47. What is your interpretation for these findings? Mention that in the text.

Page 12, lines 315-327: We have discussed the trend in patient characteristics and how the shift towards younger patients may be due to the selection of patients likely to need shorter length of stay during the pandemic compared with before. We have also contrasted this with the increase in proportion of patients having more comorbidities treated during the pandemic.

48. Please add a reference to support your results.

Page 12, lines 314-327: We have included a more detailed discussion of these findings, contrasting with previous research from the same hospital.

49. This is repeating the results, where is your interpretation for this point.

We have replaced this sentence with the first sentence from the following paragraph (see below).

50. This should be the first paragraph.

Page 11, lines 296-298: Done

51. What is your interpretation for that.

Page 11, lines 303-306: We have included a possible explanation, that revision procedures were delayed due to the potential need for intensive care units.

52. Add reference for this statement.

Page 11, line 311: We have included a reference regarding the longer length of stay for revisions compared with primaries. We are unable to find a suitable reference regarding the suitability of private hospitals for revisions and have therefore removed that part of the sentence.

53. Is there any relationship between this statement and pandemic?

The reduction in volume of revision operations is likely to be the result of the pandemic, however the specific concern of this reduction to Trusts is due to the significant impact of delays to revision operations rather than the pandemic being the cause of those delays. Earlier in the paragraph we highlighted the importance of avoiding significant delays to revisions.

54. This should be in the end of the conclusions.

Page 13, lines 368-370: Done

55. Change forecast to: predicted throughout the manuscript. Don't use two terms for the same meaning.

We have replaced all occurrences of ‘forecast’ in the manuscript to ‘predict/predicted’ as appropriate

56. It is not clear, what is the shadow waves? Where is the footnote of the graph?

Figure 4: We have included an explanation in the footnote that the shaded area signifies the range from lower to upper quartiles of LOS.

57. Remove this table, everything in this table is in table 2 (in detail).

We have moved this table to the supplementary material.

58. Table 2: Please add letters to show the significant differences within the group.

We have included a description in the text of the post-hoc pairwise comparisons of the comorbidity groups (Page 11, lines 282-289). We felt that further additions to Table 2 would make it difficult to read.

---

## [Decision Letter · Decision Letter 1]

31 Oct 2023

The impact of restricted provision of publicly funded elective hip and knee joints replacement during the COVID-19 pandemic in England

PONE-D-22-21190R1

Dear Dr. Penfold,

We’re pleased to inform you that your manuscript has been judged scientifically suitable for publication and will be formally accepted for publication once it meets all outstanding technical requirements.

Kind regards,

Ka Chun Chong

Academic Editor

PLOS ONE

Additional Editor Comments (optional):

The authors have addressed the comments well.

Reviewers' comments:

Reviewer's Responses to Questions

**Comments to the Author**

1. If the authors have adequately addressed your comments raised in a previous round of review and you feel that this manuscript is now acceptable for publication, you may indicate that here to bypass the “Comments to the Author” section, enter your conflict of interest statement in the “Confidential to Editor” section, and submit your "Accept" recommendation.

Reviewer #2: All comments have been addressed

Reviewer #3: All comments have been addressed

Reviewer #4: All comments have been addressed

2. Is the manuscript technically sound, and do the data support the conclusions?

Reviewer #2: Yes

Reviewer #3: Yes

Reviewer #4: Yes

3. Has the statistical analysis been performed appropriately and rigorously? 

Reviewer #2: Yes

Reviewer #3: Yes

Reviewer #4: Yes

4. Have the authors made all data underlying the findings in their manuscript fully available?

Reviewer #2: Yes

Reviewer #3: Yes

Reviewer #4: No

5. Is the manuscript presented in an intelligible fashion and written in standard English?

Reviewer #2: Yes

Reviewer #3: Yes

Reviewer #4: Yes

6. Review Comments to the Author

Reviewer #2: I have verified that all required questions have been answered and that all responses meet formatting specifications.

Reviewer #3: The required corrections have been made and the manuscript has been improved scientifically. Authors addressed all my comments. No further comments. Thanks

Reviewer #4: The authors responded to all comments to improve the manuscript. It looks good, and it can be published.

7. PLOS authors have the option to publish the peer review history of their article (what does this mean?). If published, this will include your full peer review and any attached files.

Reviewer #2: **Yes: **Firas Rashad Al-Samarai

Reviewer #3: No

Reviewer #4: No

---

## [Editor Report · Acceptance letter]

16 Nov 2023

PONE-D-22-21190R1 

The impact of restricted provision of publicly funded elective hip and knee joints replacement during the COVID-19 pandemic in England 

Dear Dr. Penfold:

I'm pleased to inform you that your manuscript has been deemed suitable for publication in PLOS ONE. Congratulations! Your manuscript is now with our production department. 

Kind regards, 

on behalf of

Dr. Ka Chun Chong 

Academic Editor

PLOS ONE